# A Multi-Filovirus Vaccine Candidate: Co-Expression of Ebola, Sudan, and Marburg Antigens in a Single Vector

**DOI:** 10.3390/vaccines8020241

**Published:** 2020-05-21

**Authors:** Sarah Sebastian, Amy Flaxman, Kuan M. Cha, Marta Ulaszewska, Ciaran Gilbride, Hannah Sharpe, Edward Wright, Alexandra J. Spencer, Stuart Dowall, Roger Hewson, Sarah Gilbert, Teresa Lambe

**Affiliations:** 1Nuffield Department of Medicine, Jenner Institute, University of Oxford, Oxford OX3 7DQ, UK; sarah.sebastian@vaccitech.co.uk (S.S.); amy.flaxman@ndm.ox.ac.uk (A.F.); kuanmcha@gmail.com (K.M.C.); marta.ulaszewska@ndm.ox.ac.uk (M.U.); ciaran.gilbride@st-annes.ox.ac.uk (C.G.); hannah.sharpe@seh.ox.ac.uk (H.S.); alex.spencer@ndm.ox.ac.uk (A.J.S.); sarah.gilbert@ndm.ox.ac.uk (S.G.); 2Current address: Vaccitech Ltd., Oxford Science Park, Oxford OX4 4GE, UK; 3School of Life Sciences, University of Sussex, Falmer BN1 9QG, UK; ew323@sussex.ac.uk; 4Public Health England, Porton Down, Salisbury, Wiltshire SP4 0JG, UK; stuart.dowall@phe.gov.uk (S.D.); roger.hewson@phe.gov.uk (R.H.)

**Keywords:** Ebola, Marburg, filovirus, vaccine, viral vector

## Abstract

In the infectious diseases field, protective immunity against individual virus species or strains does not always confer cross-reactive immunity to closely related viruses, leaving individuals susceptible to disease after exposure to related virus species. This is a significant hurdle in the field of vaccine development, in which broadly protective vaccines represent an unmet need. This is particularly evident for filoviruses, as there are multiple family members that can cause lethal haemorrhagic fever, including Zaire ebolavirus, Sudan ebolavirus, and Marburg virus. In an attempt to address this need, both pre-clinical and clinical studies previously used mixed or co-administered monovalent vaccines to prevent filovirus mediated disease. However, these multi-vaccine and multi-dose vaccination regimens do not represent a practical immunisation scheme when considering the target endemic areas. We describe here the development of a single multi-pathogen filovirus vaccine candidate based on a replication-deficient simian adenoviral vector. Our vaccine candidate encodes three different filovirus glycoproteins in one vector and induces strong cellular and humoral immunity to all three viral glycoproteins after a single vaccination. Crucially, it was found to be protective in a stringent Zaire ebolavirus challenge in guinea pigs in a one-shot vaccination regimen. This trivalent filovirus vaccine offers a tenable vaccine product that could be rapidly translated to the clinic to prevent filovirus-mediated viral haemorrhagic fever.

## 1. Introduction

Ebola virus disease (EVD) continues to cause sporadic and unpredictable outbreaks including the current outbreak in the Democratic Republic of the Congo (DRC), which has over 3400 cases to date [1]. Since 1976, there are 10 recorded outbreaks in the DRC alone. The 2013–2016 Ebola virus Zaire (EBOV) epidemic in western Africa was associated with 11,310 case fatalities [2], a figure thought to be largely underestimated. In isolated and limited outbreaks, EVD may be contained through surveillance, case tracking, and contact tracing. However, the movement of disease into densely populated areas, as in the 2013–2016 outbreak, often sees these containment measures fail. There is, therefore, a clear need for alternative means of control, such as affordable therapeutics and vaccines that can prevent EVD.

The Filovirus family consists of six widely recognised Ebolavirus species, with EBOV and Sudan ebolavirus (SUDV) being responsible for the majority of outbreaks in Africa. There is warranted concern that Marburg virus, another member of the filovirus family, may also cause serious disease outbreaks, with a recognition that cases may be under-reported as with other viral haemorrhagic fevers [3,4]. Cellular immunity was demonstrated to have a protective role during both Ebolavirus and Marburg virus (MARV) disease, and it was shown to be critically important in vaccine-mediated protection in macaques [5,6], although cross-strain cellular immune responses are difficult to identify. Neutralising and non-neutralising antibodies were also demonstrated to be critically important in non-human primate (NHP) vaccine and challenge studies. While monotherapy with a neutralising antibody was shown to be protective against Ebola virus infection in macaques [7], it is not clear that neutralising antibodies are an absolute requirement for protection, as Fc-mediated clearance and killing of infected cells seemed to be the main mechanism of protection after exposure in other studies [8]. A limited number of clinical studies reported the isolation of heterotypic ebolavirus-reactive antibodies which can recognise both EBOV and SUDV viruses. These cross-strain antibodies are rare and are typically identified in detailed mapping studies following high-titre virus exposure [9]. A small number of monoclonal antibodies were described against Marburg virus [10]. Detailed antibody mapping studies against Ebola virus and Marburg virus glycoproteins suggest these proteins are antigenically distinct, implying that a single monovalent vaccine will not be able to cross-protect against all filovirus family members.

The glycoprotein located on the virion surface is the principal antigen that was targeted in the vaccines tested during the 2013–2016 Ebola virus outbreak. An unprecedented number of clinical trials were initiated and run during this epidemic, and the feasibility of vaccination with several viral vector-based vaccines during an outbreak was demonstrated (e.g., the vesicular stomatitis virus (VSV) [11,12] and Chimpanzee-derived Adenovirus (ChAd) platforms [13,14,15,16,17,18]). However, to induce immunity against divergent filoviruses, either a mixture of single-pathogen vaccines or a single vaccine that encodes antigens from multiple filoviruses (a multi-pathogen vaccine) is needed. A number of studies explored the feasibility of mixing vaccines against filovirus family members with encouraging results, including clinical trials mixing two adenoviruses that encode EBOV and SUDV antigens in separate vaccine backbones [19,20]. However, as each vaccine must be manufactured individually before generating a mixed product, costs for such a combination vaccine will be significantly higher than the cost of a single multi-pathogen vaccine. With regard to filoviruses, two such single-vector multi-pathogen vaccine candidates were developed to date, based on different viral vector platforms (VSV [21] and MVA-BN-Filo [22]). Both of these showed efficacy in animal models against all targeted filoviruses, and the MVA-BN-Filo candidate was assessed in several clinical trials (up to Phase III) as a boosting vaccination after an adenoviral-vectored prime [15,17,18]. Major considerations in the development of such a multi-pathogen vaccine are protective efficacy after a single administration (even for a limited duration), scalability of manufacture to millions of doses, and an excellent safety profile. Current single-vector vaccine candidates do not satisfy all of these desired characteristics. Specifically, MVA-BN-Filo is not sufficiently immunogenic after a single shot and must be used in the context of a heterologous prime/boost regimen [15], while VSV-based candidates are live, replication-competent vectors with undesirable adverse effects such as high-grade fevers, arthritis, and rash in a minority of subjects [23,24,25,26].

Therefore, bearing these real-life constraints in mind, we sought to develop a single-vector multi-pathogen filovirus vaccine, which is based on a scalable and safe vaccine platform. Our vector of choice, the replication-deficient chimpanzee-derived adenovirus ChAdOx1, was used in 10 clinical trials to date with excellent safety and immunogenicity profiles [27]. Crucially, its cargo capacity for vaccine antigens is relatively large (up to 7 kb), compared to other viral vectors, which allowed us to construct vectors encoding three filovirus antigens at once. Briefly, we firstly generated a panel of five different triFilo vaccine vectors, each encoding three filovirus glycoproteins (EBOV, SUDV, MARV) but differing in the type and arrangement of antigen cassettes inserted into the vector backbone. After assessing expression of the encoded antigens by Western blot, we tested the immunogenic potential of these vectors in single-vaccination regimens in mice. The most promising vector was then taken forward and assessed in a lethal Zaire ebolavirus challenge model in guinea pigs, where it showed protective efficacy after a single immunisation.

## 2. Materials and Methods

### 2.1. Ethics Statement

All animal procedures were conducted in accordance with the Animal (Scientific Procedures) Act 1986. All mouse procedures were carried out under United Kingdom (UK) Home Office Project licences 30/2889 and P9804B4F1, were approved by the University of Oxford Animal Care and Ethical Review Committee, and were carried out at the University of Oxford, Old Road Campus. All guinea pig procedures were carried out under UK Home Office Project licence P82D9CB4B, were approved by the Public Health England (PHE) Animal Welfare Ethics Review Board (AWERB), and were carried out at PHE, Porton Down, Salisbury. All procedures involving infectious EBOV were performed in the Containment Level (CL) 4 facility at Public Health England (PHE) following standard laboratory procedures and risk assessments. All animal work was carried out in accordance with the UK Home Office Animal testing and research Guidance as per the Animals (Scientific Procedures) Act 1986.

### 2.2. Antigens

The following amino-acid sequences were used in vector construction: EBOV (Ebola Zaire virus glycoprotein, Makona-Kissidougou-C15 GenBank: KJ660346.1), SUDV (Sudan virus glycoprotein, UniProtKB Q66814.1), and MARV (Marburg virus glycoprotein, UniProtKB Q1PD50.1). Antigen sequences were synthesised by GeneArt (ThermoFisher, Germany) after codon optimisation for *Homo sapiens*.

### 2.3. Vector Construction

The ChAdOx1 vector was derived as previously published [28]. To generate recombinant vectors, antigen cassettes consisting of a TetR-repressible cytomegalovirus (CMV) immediate early promoter, antigen-coding sequence, and polyA sequence were inserted into the viral backbone using the Gateway recombination system (Life Technologies). Briefly, antigen cassettes were cloned into an ENTRY plasmid, sequence-verified, and recombined in vitro with ChAdOx1-DEST or ChAdOx1-biDEST (containing a Gateway destination cassette in the E1 locus, or in both the E1 and E4 loci, respectively). E1 and E4 expression cassettes both contained the TetR-repressible CMV promoter; E1 cassettes contained the BGH polyA sequence, while E4 expression cassettes contained the SV40 polyA sequence. E1 insertion occurs at the deleted E1 locus, while the E4 insertion site is located upstream of the intact E4 region. Monovalent control vectors encode the vaccine antigen at the E1 locus.

The filovirus antigens in the triFilo (2A) vector are separated by 2A ribosomal skipping sequences from porcine teschovirus-1 (P2A) and Thosea asigna virus (T2A) [29], resulting in the expression of three separate glycoproteins. The filovirus antigens in the triFilo(gly) vector are separated by GGGSGGG linkers, resulting in the generation of a single large polypeptide. TriFilo(biE1) encodes a bidirectional CMV promoter, derived from pBI-CMV1 (Clontech), modified by insertion of TetO sequences, driving SUDV and EBOV in the E1 locus, while MARV is encoded at the E4 locus. TriFilo(tandem-E4) encodes SUDV and EBOV cassettes arranged in tandem at the E4 locus and MARV at the E1 locus, while triFilo(biE4) encodes a bidirectional CMV promoter driving SUDV and EBOV at the E4 locus, and MARV at the E1 locus.

The recombinant MVA vector was constructed using bacterial artificial chromosome (BAC) recombineering methods as described previously [30]. EBOV and SUDV glycoprotein coding sequences (under the control of the short synthetic promoter (SSP) and the modified H5 (mH5) promoter, respectively) were inserted at the F11 locus, and MARV glycoprotein coding sequence (under the control of the short synthetic promoter) was inserted at the B8 locus. This multivalent MVA additionally contains an expression cassette for the Lassa virus glycoprotein, which is not mentioned in the main text, as this antigen was not assessed in the context of this project.

### 2.4. Virus Production and Expression Testing

Viral vectors were produced at the Viral Vector Core Facility at the Jenner Institute using standard methods [28,31]. All adenovirus vectors were produced in the T-REx-293 cell line (Thermo Fisher Scientific), which allows for transcriptional repression of the vaccine antigens during vector production. Vectors underwent quality control (including titration, identity PCR and sterility testing) before being used in in vitro and in vivo studies.

Expression of vaccine antigens from monovalent, bivalent, and multivalent viral vectors was assessed by western blot according to standard methods. Briefly, HEK293 cells were infected with vectors (multiplicity of infection(MOI) = 1), harvested after 24 h, and lysed in lysis buffer. Reduced and denatured lysates were resolved by 4%–12% SDS-PAGE and transferred onto nitrocellulose. Glycoproteins were detected using mouse antiserum from mice previously vaccinated with monovalent vectors (ChAdOx1-EBOV, ChAdOx1-SUDV) or commercial anti-MARV glycoprotein (GP) antibody (ab190459, abcam), horseradish peroxidase (HRP)-conjugated secondary antibodies, and chemiluminescence imaging (ChemiDoc, BioRad, Hercules, CA, USA).

### 2.5. Mouse Experiment Design

Six-week-old female BALB/c mice were obtained from Envigo (Blackthorn, UK). Six-week-old CD-1 mice were obtained from Charles River (Harlow, UK). On arrival, animals were randomly distributed into individually ventilated cages, housed in groups of three, four, five, or six under specific pathogen-free conditions, and fed and watered *ad libitum* with a 12-h/12-h light/dark cycle. After seven days of settling in, mice were anesthetised using vaporised IsoFlo^®^ and vaccinated intramuscularly (i.m.) with 50-µL doses of 10^8^ infectious units (IU) ChAdOx1 in PBS. Blood samples were taken from the tail vein. In prime-boost experiments, booster vaccinations of 10^6^ plaque-forming units (PFU) MVA in PBS were administered after the relevant time interval. Mice were culled humanely at the end point of the experiment via an approved Schedule 1 method; cardiac blood and spleens were harvested for further immunological analysis. Numbers of mice per experimental group were *n* = 4 or 5 for inbred BALB/c mice and *n* = 10 for CD-1 mice, to account for higher variability in immune responses in these outbred mice.

### 2.6. ELISpot

Murine IFN-γ-producing splenocytes were assessed by ELISpot assay after vaccination with filovirus viral vectors as previously described [32], with the following exceptions: splenocytes were added to ELISpot plates at concentrations varying from 1.25 × 10^5^ to 5 × 10^5^ cells/well and stimulated with pools of peptides at a final concentration of 1 μg/mL per peptide. Peptide pools consisted of 15-mer peptides overlapping by 11 amino acids, spanning EBOV GP, SUDV GP, or MARV GP. For graphical presentation, the number of IFN-γ-producing cells was calculated as the number of spot-forming cells in the presence of peptides minus the number of spot-forming cells without peptides.

### 2.7. ELISA

Antibody responses were measured against trimerised EBOV GP (amino acids 1–649 of GenBank protein AHX24649.1, with a C-tag), produced in house as described previously [13]. Antibody responses against monomeric SUDV GP (made in house) and recombinant MARV-Angola GP (Alpha Diagnostic International) were also measured. Reference pools of each of EBOV GP, SUDV GP, and MARV GP antibody-positive mouse sera were used to form a standard curve for each plate. The relevant pool was added at an initial dilution of 1:250 (EBOV GP or MARV GP) or 1:125 (SUDV GP) in PBS/T and underwent 10 two-fold dilutions. An arbitrary number of ELISA units were assigned to the reference pool (62.5 AU for EBOV GP or MARV GP; 125 AU for SUDV GP), and OD values of each dilution were fitted to a four-parameter logistic curve using SOFTmax PRO software. ELISA units were calculated for each sample using the OD values of the sample and the parameters of the standard curve. All ELISA data presented are in AU.

### 2.8. Intracellular Cytokine Staining (ICS)

Splenocytes were prepared as described above, plated in 96-well round-bottom plates, and stimulated using peptide pools for EBOV GP, SUDV GP, or MARV GP (as described above) at a final concentration of 5 µg/mL or media only. Stimulation and staining was then performed as described previously [33] except that the following antibodies were used: anti-CD4-Qdot605, anti-CD127-APCef780 (Invitrogen), anti-CD62L-PeCy7, and anti-CD8-PerCP/Cy5.5 antibodies (eBioscience), as well as LIVE/DEAD^®^ Fixable Aqua Dead Cell Stain Kit (Thermo Fisher Scientific), anti-TNF-Alexa488, anti-IL-2-PE, and anti-IFN-γ-e450 antibodies (eBioscience). Antigen-specific cells were identified by gating based on doublet negative, size, live cells, and either CD4^+^ or CD8^+^ surface expression. Background responses in unstimulated control samples were subtracted from responses of peptide stimulated T cells.

### 2.9. Neutralising Antibody Titres

Neutralising antibodies were measured using pseudotyped lentiviruses, produced as described previously [34], expressing either the glycoprotein from Zaire ebolavirus Makona isolate (*H. sapiens*-wt/GIN/2014/Kissidougou-C15; KJ660346.1), Sudan ebolavirus (Boniface/SUD/1976; FJ968794), or Marburg marburgvirus Angola isolate (Ang0998; DQ447660).The assays were run in duplicate using a virus input of 100 × TCID_50_ and reciprocal serum dilution range of 1:20–1:640. The blocking ability of vaccine-induced antibodies was assessed with a readout of the 50% inhibitory concentration (IC_50_).

### 2.10. Statistics

Statistical analyses were performed using GraphPad Prism version 7.01. Grouped data are presented as means with standard error of the mean (SEM), unless otherwise indicated. Statistical significance of variations in continuous variables by group was analysed by Mann–Whitney or Kruskal–Wallis tests (for skewed data) or *t*-tests or ANOVA (for normally distributed data) as stated in the results. For comparisons across multiple groups, Dunn’s multiple comparisons test was used for skewed data, and the Holm–Sidak multiple comparisons test was used for normally distributed data. These corrections for multiple comparisons were recommended by GraphPad Prism.

### 2.11. Guinea Pig Experiment Design

Groups of female Hartley strain guinea pigs (*n* = 6/group) were intra-muscularly vaccinated with 5 × 10^8^ IU of ChAdOx1-triFilo(2A) or a mix of monovalent ChAdOx1 controls (ChAdOx1-EBOV, ChAdOx1-SUDV, and ChAdOx1-MARV) or a negative control (ChAdOx1 with irrelevant antigen). Then, 28 days after immunisation, the vaccinated animals were challenged subcutaneously with a lethal dose (10^3^ TCID_50_) of guinea pig-adapted EBOV (EBOV Yambuku-Ecran strain [35]). The EBOV was passaged five times in guinea pigs to achieve lethality, as previously described [36]. Virus was titrated by 50% tissue culture infective dose (TCID_50_) assay in VeroE6 cells (European Collection of Cell Cultures, UK). Animals were assessed daily with respect to temperature and weight loss throughout the experiment. Clinical signs were monitored at least twice daily, and the following numerical score was assigned for analysis: 0 (normal); 2 (ruffled fur); 3 (lethargy, hunched, and wasp-waisted); 5 (rapid breathing); 10 (immobile, neurological). To prevent unnecessary suffering to animals, humane clinical endpoints were used where animals would be culled upon reaching 10% weight loss and a moderate clinical sign or any of the following: 20% weight loss; immobility; paralysis; neurological signs.

### 2.12. Viral Loads and Histology

Tissue samples were weighed and homogenised. Total RNA was extracted using the RNeasy Mini Kit (Qiagen, UK) and eluted in 50 μL of RNase-free water. A Zaire ebolavirus strain-specific real-time RT-PCR assay was utilised for the detection of viral RNA with primer and probe sequences adopted from a published method [37]. Real-time RT-PCR was performed using the SuperScript III Platinum One-step qRT-PCR kit (Life Technologies, Carlsbad, CA, USA). Reactions were run and analysed on the QuantStudio Real-Time Platform (Life Technologies) using software version 1.2. Quantification of viral load in samples was performed using a dilution series of quantified RNA oligonucleotide (Integrated DNA Technologies). For histology, tissue samples were processed to paraffin wax; sections were cut at approximately 3–5 µm thick, stained with haematoxylin and eosin (HE), and examined microscopically. In addition, sections from each animal were stained for Ebola viral RNA using the Leica BondMax (Leica Biosystems) and the Leica Bond Polymer Refine Red Detection Kit (Leica Biosystems). An antigen retrieval step was included for 10 min using the Bond Enzyme Pre-Treatment Kit, Enzyme 3 (three drops). A rabbit polyclonal, anti-Ebola virus antibody (IBT Bioservices) (dilution 1:2000) was incubated with the slides for 60 min. Alkaline phosphatase and haematoxylin counterstains were used to visualise the slide. Appropriate positive and negative tissue and reagent controls were included.

## 3. Results

### 3.1. Vaccine Design

Expression of two or more antigens from a single adenoviral vaccine vector can be challenging, and optimisation of expression cassettes, directionality, and insertion loci may be necessary [38]. Antigens can either be linked (and expressed from a single transcript) or cloned as separate expression cassettes into several well-characterised insertion sites (e.g., E1, E3, E4). We, therefore, firstly constructed a panel of triFilo adenoviral vectors encoding varying constellations of the glycoproteins from the three filovirus family members most likely to cause a severe viral haemorrhagic fever (VHF) outbreak: Zaire ebolavirus (EBOV), Sudan ebolavirus (SUDV), and Marburg virus (MARV) (Figure 1a).

Western blot analysis was performed to assess expression of the three antigens in cells infected with the respective triFilo vectors (Figure 1b). In the first iteration of viral vectors (1–3, Figure 1a), proteins of the correct size for all three inserts were detected for triFilo(2A), a construct with a ribosomal “skipping” 2A sequence between each of the antigens. In comparison, separate glycoproteins would not be expected for triFilo(gly) as this vector expresses all three glycoproteins fused together with flexible glycine linkers as one large polyprotein. The faint bands seen for this construct in SUDV GP and MARV GP Western blots indicate that some proteolytic processing may still take place. For the third construct, triFilo(biE1), only expression of a correctly sized SUDV GP was detected.

Two alternative vectors (triFilo(tandem-E4) and triFilo(bi-E4)) were generated in an attempt to maximise protein expression from individual antigen cassettes (4–5, Figure 1a). Expression of a correctly sized MARV glycoprotein was detected from both triFilo(tandem-E4) and triFilo(bi-E4) vectors. However, only the triFilo(tandem-E4) expressed EBOV (weakly) and SUDV glycoproteins (Figure 1b).

### 3.2. Cellular Immunogenicity

It was previously shown that combining liver- and blood-stage malaria viral-vectored vaccines results in CD8^+^ T-cell interference [39,40], and, as the cellular immune response was shown to play an important role in protection against filovirus-mediated disease, we assessed cellular immunogenicity by ELISpot following a single vaccination with our triFilo vectors in BALB/c mice. Mice in the control group were vaccinated with a mixture of monovalent vectors at a final viral infectious unit dose comparable to the trivalent vaccine regimen. Immune responses at two weeks post vaccination of BALB/c mice with 1 × 10^8^ infectious units (IU) against all three antigens (EBOV GP, SUDV GP, and MARV GP) were comparable for triFilo(2A), triFilo(gly), and monovalent mix. However, responses to EBOV GP and MARV GP peptides were significantly lower (*p* = 0.009, *p* = 0.004, respectively, Dunn’s multiple comparison test) post triFilo(biE1) vaccination, compared to monovalent mix (Figure 2a). Of the alternative vaccines, poor antigen expression by western blot eliminated the triFilo(biE4) vector, resulting in only the triFilo(tandem-E4) vector progressing to immunogenicity assessment. A significantly reduced ELISpot result against EBOV GP was measured post vaccination with triFilo(tandemE4) compared to the mix of monovalent vectors (*p* = 0.08, Mann–Whitney test, Figure 2b).

To address a potential bias in our mixed controls, we assessed the impact of increasing the dose of the monovalent mix vaccine controls three-fold (to a final dose of 3 × 10^8^ IU), so that each component of the mix was equal in infectious units to the trivalent vaccine dosage (1 × 10^8^ IU). Importantly, comparable cellular immunogenicity for all three antigens was observed, whether the high- or low-dose of monovalent mix was used (Appendix A), suggesting that a three-fold difference in dose does not affect cellular immunogenicity. We also determined cellular cross-reactivity using individual monovalent vaccines expressing EBOV, SUDV, or MARV GP. We show that splenocytes from mice vaccinated with monovalent EBOV respond to a small degree to stimulation with SUDV GP peptides, but no other cross reactivity was seen (Appendix A).

### 3.3. Humoral Immunogenicity

Based on the results from the cellular immunogenicity profiling, humoral immunity after a single vaccination was determined for the two most promising vaccine candidates (triFilo(2A) and triFilo(gly)). We assessed immunoglobulin G (IgG) titres against EBOV GP, SUDV GP, and MARV GP in both inbred (BALB/c) and outbred (CD-1) mice at 10 and eight weeks post vaccination, respectively (Figure 3a,b). IgG titres against EBOV GP and SUDV GP were lower following vaccination with triFilo(gly) than those after vaccination with triFilo(2A) and monovalent mix in BALB/c (Figure 3a) and significantly so in CD-1 mice (Figure 3b; *p* = 0.008 for EBOV GP and *p* = 0.006 for SUDV GP, Dunn’s multiple comparison test).

Mice vaccinated with an irrelevant antigen did not have detectable IgG against the filovirus antigens. Additionally, in BALB/c mice, we found that IgG titres against EBOV GP and SUDV GP were comparable at 21 weeks to those at 10 weeks for each of the three vaccination regimens. Neutralising antibody titres as assessed through pseudotype neutralisation assays against EBOV and MARV were induced after triFilo(2A) vaccination, but could not be detected against SUDV (Appendix A). These humoral immunogenicity results allowed us to select the most promising candidate vaccine, triFilo(2A), for further immune profiling. We summarise the data comparing our trivalent adenoviral vaccine candidates in Table 1.

### 3.4. Prime-Boost Immune Profiling

The immune responses following a single-shot viral-vectored EBOV vaccine were protective in a field trial [41]. However, it was demonstrated previously that a booster vaccination can augment both the longevity and magnitude of immune responses, which will be important for first-in-field responders [42]. Modified Vaccinia virus Ankara (MVA) is a safe and highly immunogenic viral vaccine vector with a very strong boosting capacity that was demonstrated in multiple clinical trials [43]. We therefore generated a multivalent MVA vector expressing EBOV, SUDV, and MARV glycoproteins (multiMVA), to be used as a booster after priming mice with ChAdOx1-triFilo(2A).

We assessed the immunogenicity of this multiMVA and showed that, after a single-dose ELISpot, responses were comparable to those elicited by a monovalent mix of MVAs (Appendix A). We next assessed the boosting ability of multiMVA after triFilo(2A) prime. As expected, a robust immune response was measured after a prime-boost regimen, with different intervals between vaccines, and in different mouse strains, both inbred (BALB/c) and outbred (CD-1) (Figure 4). The response against all three filovirus antigens was comparable to a mixture of monovalent controls which were also boosted with multiMVA (Figure 4a,b). In Figure 4c, we compared prime-boost results to previous experiments (from Figure 2 for BALB/c and Appendix A, for CD-1) measuring responses after prime only. Boosting was stronger in BALB/c mice than CD-1 mice, for all three antigens; this may have been affected by both the genetic background and the interval between prime and boost (10 weeks in BALB/c mice and four weeks in CD-1 mice). A prime-boost regimen induced approximately two-fold higher responses against EBOV and SUDV GP antigens and four-fold higher against MARV GP antigen, compared to prime only, in BALB/c mice.

Intracellular cytokine analysis revealed that the dominant cellular response observed post boost was driven by IFN-γ^+^ or IFN-γ^+^ and TNF-α^+^ CD8^+^ T cells after peptide-specific stimulation which was previously associated with protective efficacy in challenge experiments in macaques [44,45]. The strongest CD8^+^ T response measured was against the SUDV glycoprotein (Figure 5). This reflects the results seen in the ELISpot assay (Figure 4).

The humoral response after triFilo(2A) priming was also increased after the MVA booster, approximately eight-fold for EBOV GP, two-fold SUDV GP, and nine-fold for MARV GP after 10 weeks in BALB/c and two-fold for EBOV GP, two-fold SUDV GP, and five-fold for MARV GP after four weeks in CD-1 mice (Appendix A). Neutralising antibody titres as assessed through pseudotype neutralisation assays against EBOV and MARV were induced after triFilo(2A) prime followed by MVA boost in both CD-1 and BALB/c mice (Appendix A). Titres against SUDV were at the lower limit of detection. It is unclear why neutralising titres against SUDV were low or undetectable in the pseudotype assay for both prime and prime-boost regimens (Appendix A), and this warrants further exploration. There were no significant differences in the neutralising antibody titre against EBOV, SUDV, or MARV in mice receiving triFilo(2A) or mono mix as their prime vaccination. These data demonstrate that the immune response against filovirus antigens can be augmented in a prime-boost regimen with viral vectored vaccines, and that a prime-boost regime induces antibodies with the ability to block viral entry.

### 3.5. Lethal EBOV Challenge

Since we observed robust immunogenicity in mice with triFilo(2A), we next assessed efficacy of this vaccine in a stringent Ebola challenge in guinea pigs.

Groups of guinea pigs (*n* = 6) were vaccinated intramuscularly with a final concentration of 5 × 10^8^ IU of ChAdOx1 triFilo(2A), a mix of relevant monovalent ChAdOx1 controls, or a negative ChAdOx1 control with irrelevant antigen. The vaccinated animals were challenged intraperitoneally with 10^3^ TCID_50_ EBOV (using an EBOV challenge virus derived from the Mayinga Zaire strain [37]) 28 days after immunisation. Vaccination with the ChAdOx1 vector did not result in adverse effects (Appendix A).

After challenge with EBOV, guinea pigs who were vaccinated with the control ChAdOx1 vaccine lost weight from day 5 onwards (Figure 6b), alongside an increase in body temperatures (Figure 6c). Animals vaccinated with ChAdOx1-triFilo(2A) or a mix of monovalent control ChAdOx1 vaccines continued to put on weight post challenge and showed no significant fluctuations in temperatures (Figure 6b,c).

Throughout the course of the study, animals were assessed for clinical signs, and a numerical score for each clinical sign was recorded. Results showed that clinical signs were first observed in the ChAdOx1 negative-control group on day 5 post challenge (Appendix A). None of the animals vaccinated with ChAdOx1-triFilo(2A) or a mix of monovalent control vaccines exhibited any clinical signs and scored normal throughout the course of the study (Appendix A). These animals all survived until the scheduled end of the study (21 days post challenge), whereas animals receiving the negative-control vaccine all met humane endpoints by day 9 post challenge (Figure 6d).

At necropsy, samples of blood, liver, and spleen were assessed for local viral loads by PCR. High levels of EBOV genomes were found in the ChAdOx1 negative-control group in all of the tissues tested, while those animals that received ChAdOx1-triFilo(2A) or a mix of monovalent control vaccines remained below the limit of detection of the assay (Figure 6e).

Vaccination with ChAdOx1 triFilo(2A) or a mix of monovalent control vaccines appeared to protect against lesions attributable to EBOV infection as assessed through histology; this correlates with all animals in the test groups surviving to the study endpoint (Appendix A).

## 4. Discussion

The humanitarian need for a prophylactic intervention against Ebola virus was highlighted by the 2013–2016 Zaire ebolavirus outbreak and the reoccurring outbreaks of the virus, including the current outbreak in the Democratic Republic of Congo [1]. To date, filovirus vaccine candidates progressed to Phase I or further have been predominantly viral vector-based. The only vaccine to be assessed for efficacy in the 2013 outbreak is an attenuated vesicular stomatitis virus vector encoding the glycoprotein from an ancestral Ebola virus strain, rVSV-ZEBOV [46], which demonstrated 100% efficacy in a Phase III ring vaccination trial [11] and is being used in the current DRC outbreak. The rVSV-ZEBOV vaccine is associated with a number of adverse effects such as high-grade temperatures and vaccine-induced arthritis, dermatitis, and vasculitis with approximately 23% of participants in the phase III efficacy clinical trial reporting either arthritis and/or arthralgia [23,24,25,26]. These side-effects are not routinely seen post vaccination with adenoviral vectored vaccines [47]. The Ad prime, MVA boost regimen that was clinically assessed during the 2013–2016 Ebola outbreak is also being trialled in central Africa, and it was submitted to the FDA for licensure using the animal rule. It was demonstrated that adenoviral vectored Ebola virus vaccines induce the same level of humoral immunity as rVSV-ZEBOV and, advantageously, the adenoviral vaccines also induced strong and long-lived cellular immunity [13]. High levels of both cellular and humoral immunity were demonstrated in Ebola virus survivors, and both T cells and antibodies were demonstrated to play a key role in protection against Zaire ebolavirus challenge in NHP [48,49,50,51]. This underlines the importance of developing a vaccine that can induce both humoral and cellular immune responses.

The preferred target product profile recommended by the WHO describes a vaccine that targets multiple filovirus species; unfortunately, the majority of vaccines currently in advanced clinical testing do not meet this criterion. Indeed, only a limited number of these platforms can simultaneously encode or contain multiple disease antigens and concurrently induce strong humoral and cellular immunogenicity against each antigen without immune competition. The aim of this study was to generate a single adenoviral vaccine vector that could provide protection against disease caused by three different filovirus family members. Adenoviral vectors can be designed to express antigens from multiple independent expression cassettes (each containing its own promoter and polyA sequence) or from one single cassette, with antigens linked by short elements such as glycine linkers, 2A ribosomal skipping sequences, or internal ribosome entry sequences (IRES). We chose to generate a representative selection of triFilo vectors predominantly focusing on constructs that contained three individual expression cassettes, based on the reasoning that these would most likely result in the expression of three correctly folded glycoprotein antigens. Perhaps surprisingly, therefore, it was the vector in which the three antigens were linked with 2A sequences (triFilo(2A)) that prevailed, both in expression studies and in immunogenicity and efficacy assessments (see summary of findings in Table 1). The EBOV challenge in guinea pigs allowed us to indirectly compare our vaccine with a lead clinical EBOV vaccine strategy: a monovalent adenovirus expressing EBOV glycoprotein. Encouragingly, our multivalent vaccine performed as well as a monovalent adenovirus expressing EBOV glycoprotein only. 

In order to confirm that ChAdOx1-triFilo(2A) merits further (clinical) development, several questions remain to be addressed. Firstly, genomic stability of the vector should be assessed through repetitive passaging of the vector in adenoviral producer cells to ensure suitability for large-scale GMP manufacture. Glycoprotein antigens can be cytotoxic when expressed at high levels; thus, spontaneous mutations in the vector that downregulate antigen expression during vector production can confer a selective advantage, creating vector subpopulations that may outgrow the correct clone during large-scale production. To mitigate against this, we routinely employ a TetR-based system to prevent antigen expression during vector production, and genomic stability testing is performed in a TetR-producer cell line. Secondly, a limitation of the work presented here is the lack of challenge studies with regard to SUDV and MARV, as well as other EBOV strains. Protective efficacy of our vector was so far shown in the context of infection with EBOV. The profile of our triFilo(2A) candidate vaccine could be further strengthened by efficacy studies with the other vaccine targets including SUDV and MARV, as well as the quantification of neutralising antibodies against other filoviruses. In order to move this candidate into the clinical development phase, efficacy assessments in NHP against all three vaccine targets (EBOV, SUDV, MARV) would also be desirable. Thirdly, it may be beneficial to develop this vaccine candidate as part of a prime-boost combination, in order to augment and prolong the vaccine-induced immune response. In this study, we showed that an MVA vector encoding the same three filovirus antigens can act as a potent boosting agent. MVA has an excellent safety profile and it has been assessed extensively in the young (including babies), older adults, and immunocompromised individuals. Indeed, Bavarian Nordic developed MVA-BN-Filo, which encodes the glycoproteins of EBOV, SUDV, and MARV and the nucleoprotein from Tai forest virus. This vaccine was assessed in heterologous prime-boost together with Ad26-ZEBOV [17,18] with promising results and long-lived immunity, and this regimen advanced to application for licensure approval. Our multivalent adenovirus adds value to the field as it encodes three filovirus antigens, conferring a broader immune response than the current Ad26 platform.

## 5. Conclusions

We describe here our multivalent adenoviral-vectored vaccine candidate expressing glycoproteins from Zaire ebolavirus, Sudan ebolavirus, and Marburg virus; triFilo(2A). We demonstrate that this vaccine induces strong humoral and cellular immunity in two mouse strains. triFilo(2A) confers protective efficacy against one of the three vaccine targets in an animal challenge model. Our data provide a rationale for further development of a single-dose multivalent filovirus vaccine which could be deployed quickly and easily in outbreak situations.

## Figures and Tables

**Figure 1 vaccines-08-00241-f001:**
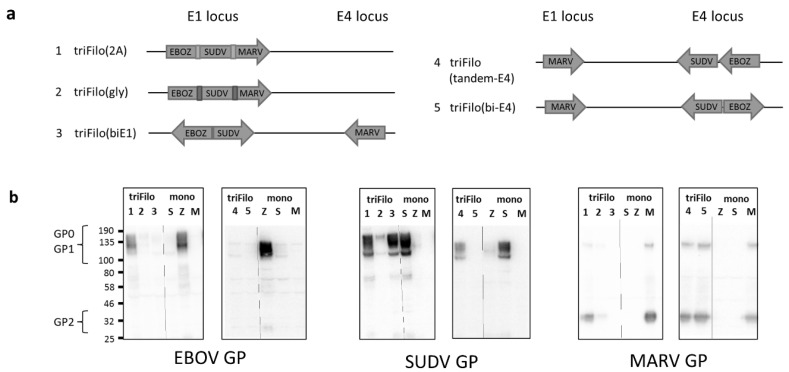
ChAdOx1 triFilo constructs. (**a**) Schematic to show design of five triFilo vectors. (**b**) Expression by western blot of each of the three antigens in the triFilo constructs compared to monovalent controls (S = Sudan ebolavirus (SUDV) glycoprotein (GP), Z = Zaire ebolavirus (EBOV) GP, M = Marburg virus (MARV) GP).

**Figure 2 vaccines-08-00241-f002:**
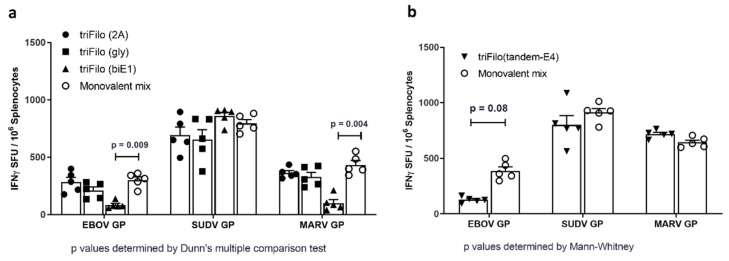
Cellular immunogenicity of ChAdOx1 triFilo vaccines. Mice (*n* = 5 per group) received either a triFilo vaccine or mix of monovalent controls. IFN-γ ELISpot on BALB/c samples two weeks post vaccination. Splenocytes were stimulated ex vivo with peptide pools spanning EBOV GP, SUDV GP, or MARV GP glycoproteins. (**a**) First-generation trivalent constructs compared to monovalent mix; *p*-values were determined by Dunn’s multiple comparison test. (**b**) Alternative trivalent construct compared to monovalent mix; *p*-values were determined by Mann–Whitney test.

**Figure 3 vaccines-08-00241-f003:**
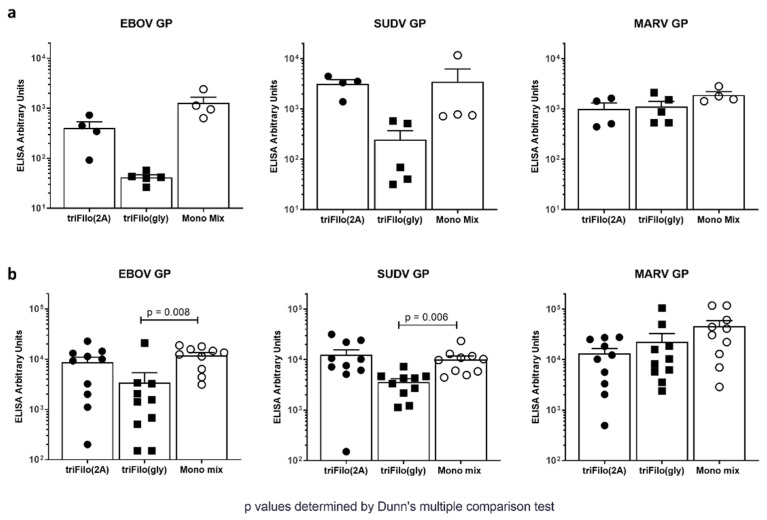
Humoral immunogenicity of ChAdOx1 triFilo vaccines. Mice received either a triFilo vaccine or mix of monovalent controls. Anti-glycoprotein ELISA (EBOV, SUDV, or MARV GP) was used to quantify immunoglobulin G (IgG) titres induced by vaccination in (**a**) BALB/c mice (*n* = 4 per group) 10 weeks post immunisation and (**b**) CD-1 mice (*n* = 10 per group) eight weeks post immunisation; *p*-values were determined by Dunn’s multiple comparison test.

**Figure 4 vaccines-08-00241-f004:**
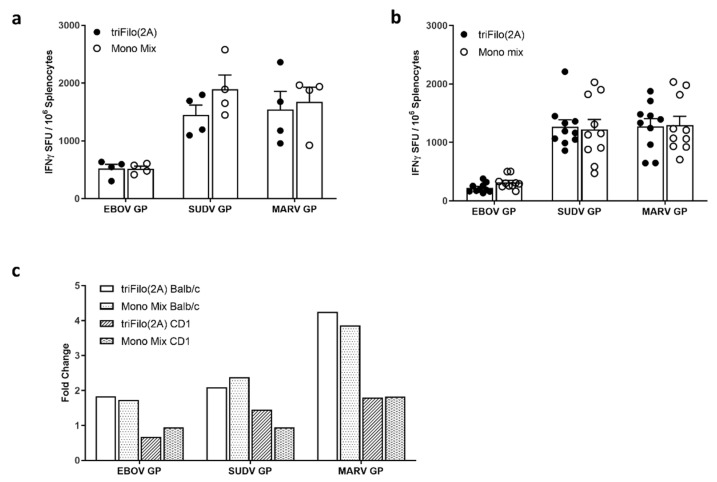
Cellular immunogenicity of ChAdOx1 triFilo(2A) prime followed by multiMVA (Modified Vaccinia virus Ankara) boost. Mice were primed with either triFilo(2A) vaccine or mix of monovalent controls, subsequently all mice were boosted with multiMVA. Splenocytes were stimulated ex vivo with peptide pools spanning EBOV, SUDV, or MARV glycoproteins. (**a**) IFN-γ ELISpot on Balb/c samples (*n* = 4 per group) 2.5 weeks post boost vaccination, with 10 weeks prime-boost interval. (**b**) IFN-γ ELISpot in CD-1 mice (*n* = 10 per group) two weeks post boost vaccination, with four weeks prime-boost interval. (**c**) Fold change of IFN-γ ELISpot in both mouse strains—two weeks prime only compared to prime boost.

**Figure 5 vaccines-08-00241-f005:**
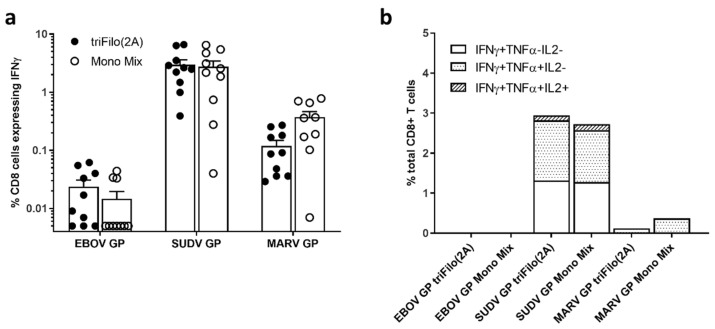
T-cell immunogenicity of prime-boost vaccination determined by intracellular cytokine staining (ICS). CD-1 mice (*n* = 10 per group) were primed with either triFilo (2A) vaccine or mix of monovalent controls; subsequently, all mice were boosted with multiMVA. Cytokine levels determined by ICS for T cells two weeks post boost vaccination, with four weeks prime-boost interval. (**a**) IFN-γ^+^ CD8^+^ T cells. (**b**) CD8^+^ T cells expressing IFN-γ, TNF-α, and IL-2.

**Figure 6 vaccines-08-00241-f006:**
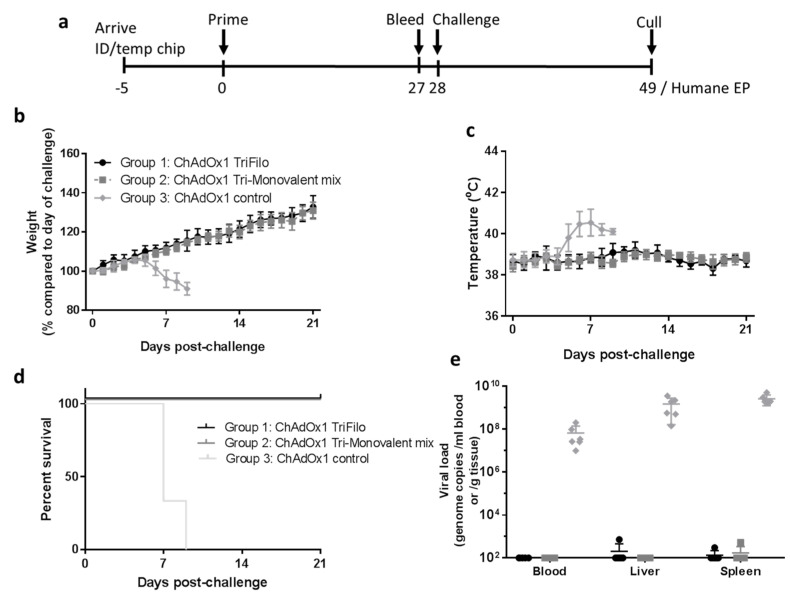
Heterologous EBOV challenge in guinea pigs. Guinea pigs (*n* = 6 per group) were vaccinated with triFilo(2A) vaccine, a mix of monovalent controls, or ChAdOx1 expressing an irrelevant antigen. Subsequently, all guinea pigs were challenged with Ebola Zaire. (**a**) Study design, (**b**) weight, and (**c**) temperatures of animals immunised with ChAdOx1 filovirus vaccines or ChAdOx1 control after challenge with Ebola virus. Graphs show mean values from up to six animals per group with error bars denoting standard error. (**d**) Kaplan–Meier survival plot of ChAdOx1 filovirus vaccine-immunised animals compared to ChAdOx1 control after challenge with Ebola virus. (**e**) Ebola viral RNA levels in tissues of animals immunised with ChAdOx1 filovirus vaccines or ChAdOx1 control after challenge with Ebola virus (10^2^ is the lower limit of detection of the assay). Graphs show values from six animals per group with lines representing mean values and error bars denoting standard error.

**Table 1 vaccines-08-00241-t001:** Summary of single-vector trivalent filovirus vaccine candidates assessed in this study. Scale: +++ denotes equivalency to relevant control, + and ++ denote less than control, − denotes undetectable; n/a = not assessed.

Vector ^#^	Vector Name		Expression by Western Blot ^#^	Antibody Immunogenicity * (BALB/c /CD-1)	T-Cell Immunogenicity * (BALB/c)
1	triFilo(2A)	EBOV	++	++/+++	+++
		SUDV	+++	+++/+++	++
		MARV	++	+++/++	+++
2	triFilo(gly)	EBOV	+	+/++	+++
		SUDV	++	++/++	++
		MARV	+	+++/+++	+++
3	triFilo (biE1)	EBOV	−	n/a	+
		SUDV	+++	n/a	+++
		MARV	−	n/a	+
4	triFilo (tandem-E4)	EBOV	+	n/a	+
		SUDV	++	n/a	+++
		MARV	+++	n/a	+++
5	triFilo(bi-E4)	EBOV	−	n/a	n/a
		SUDV	−	n/a	n/a
		MARV	+++	n/a	n/a

^#^ Compared to monovalent control vector. * Compared to a mix of monovalent vectors.

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
