# Peer review of "A Multi-Filovirus Vaccine Candidate: Co-Expression of Ebola, Sudan, and Marburg Antigens in a Single Vector"

_vaccines, 2020, doi:10.3390/vaccines8020241_

Round 1
Reviewer 1 Report
“A Multi Filovirus Vaccine: Co-Expression of Ebola, Sudan and Marburg Antigens in a Single Vector to Prevent Viral Haemorrhagic Fever”
The authors of the present study report the development and testing of a trivalent vaccine capable of expressing three filovirus glycoproteins from the same vector, streamlining production costs by reducing the need for additional vectors mixed into a vaccine to achieve the same protection against the same pathogens. The authors claim that these efficiencies can help make their vaccine candidate a more accessible option in poorer regions where community protection is of vital importance (such as in sub-Saharan Africa with its history of recurrent filovirus outbreaks). Further, they go on to test an MVA-based multivalent vector as a boost vaccine, which the authors predict could be useful for increasing vaccine efficacy for first-line responders.
While their findings are clearly presented in a well-written article that indeed offers the field a potentially useful and important vaccine candidate that may warrant pursuit in future studies, at this time there are a few notable concerns that prevent me from being able to recommend accepting the manuscript in its current form. Below are the main comments to address, as well as additional minor suggestions or editing fixes.
Major Comments
- The abstract and introduction could give the impression to some readers that a trivalent, single vector filovirus vaccine has not been previously developed. Indeed, this is not the case, as others (Mire et al, 2015; Callendret et al, 2018) have shown efficacy of such vectors. While MVA-BN-Filo is later mentioned in the discussion, an rVSV vaccine capable of protection against the same filoviruses (Mire et al, 2015) is not mentioned in the paper. In both regards, citation and discussion of these previous studies should be better developed upfront within in the introduction to more clearly illustrate a) what sets the authors’ current study apart from these and others previously conducted and b) why rVSV based vaccines might not be ideal.
- While it is mentioned as a caveat within the discussion, it is unclear why only EBOV, and only a single variant, was tested for the guinea pig survival study. Why not multiple viral strains, and especially multiple species, including MARV and SUDV? The lack of in vivo data for any other virus works to undermine the appeal of this vaccine as offering true broader protection. In this regard, the study feels like an incomplete work, and without substantial justification, a separate follow-up report to address a question that should be determined in the context of this current study seems unwarranted.
- Reference 20 appears to be a concurrent, similar manuscript, also from the same team, focusing on a Lassa GP component in place of Sudan GP in, presumably, the same vector. Given the presence of the Lassa GP cassette also within the MVA booster, it seems likely that this vector is also utilized in that work. If MVA is capable of housing four different GPs, and has been shown in previous studies to function quite well as a prime vaccine candidate and not simply as a boost vaccine, it seems that testing of MVA as a prime vaccine candidate could be worthwhile, as this would potentially offer the broadest protection available in a prime-only regimen and save even more costs. Meanwhile, triFilo(2A) or Lassa-containing ChAdOx1 vector could be tested as boosters, with these two trivalent vectors instead reserved for first line responders as they might be needed less frequently and could be better tailored to individual responder’s needs in the field. Rationale or data showing if an MVA prime strategy could be efficacious (or why it wouldn’t work) would be helpful in addressing this seemingly open question.
- Does the inclusion of the Lassa GP cassette in the MVA boost vector in any way interfere with any MVA-related results? Data addressing this (even if supplemental) would be beneficial.
- It is shown that the prime-boost strategy can induce neutralizing antibodies and augment immune responses, but this disappears from the paper, which then switches back to the prime only vaccine for the guinea pig protection study. Explain how the inclusion of the prime-boost strategy in this particular paper is relevant if prime alone seems to protect on its own and the main point of the study is that the single vector, single dose vaccine reduces costs. If both strategies can be justified in the context of this paper, then why were no neutralization or ICA assays performed for prime only? Conversely, why was no guinea pig study performed to further validate prime-boost protection efficacy? As it currently stands, the prime-boost data feels like an aside, rather than a core component of the central throughline of the paper.
- Tying into the above comment, it seems there would be more consistency if the mouse models and maEBOV were used for the protection study, rather than (or in addition to) the guinea pigs, to directly compare to the earlier study data. Alternatively, show results of similar assays with the guinea pig protection experiment (such as humoral and cellular immunogenicity, neutralization assay, etc.) to ensure that this data aligns well to the previous mice data. There seems to be a lack of consistency regarding what experiments/assays are performed on what animals and whether boost vector is involved or not.
- Explain why the two mouse strains are inoculated at different intervals between prime and boost, as well as why they are sampled at different timepoints post vaccination (particularly when it comes to CD-1, i.e. Fig S2). Further, as seen in Fig 4 data, is it fair to directly compare 2 week prime-only data, to separate prime-boost data, when the timelines are so different between these individual experiments and mouse strains? Please provide justification.
Minor Comments
The distinction between 1st and 2nd gen vectors seems unnecessary. While they might have been produced that way historically, the reader likely doesn’t need any context of what’s made first or second – only what works and what doesn’t. It could make for a more streamlined story if results are shown simply as if there are five vaccine candidates produced and of them, 2A eventually was deemed best. Also, 2nd gen implies an improvement in what came before, but that’s clearly not the case here. By making this change, Fig 2 could then be one larger, single panel figure.
The Materials section could use a bit more detail in certain key sections, particularly those that leave key information within only Reference 20. Such extra useful details include: info on primer sets and abs used for tests; clarification of how the triFilo(gly) polypeptide is cleaved; details on ICS flow cytometry; clarification of what “peptide pools” means; mentioning TNF and IL-2 staining in addition to IFNy; including a biosafety statement in relation to BSL4 based experiments; and describing briefly the normalization/standardization method for measuring genome copies.
Clarify the total #s of mice/guinea pigs per experiment and how many mice are in each group in each experiment (it’s not immediately clear that it’s 6 guinea pigs per group). Why are different animal numbers per group used between experiments?
How does gp-adapted virus differ to wild type virus? Are there any changes within EBOV GP that would be expected to affect outcomes seen in guinea pigs compared to those that might be seen when using the wild type virus in a susceptible animal?
Line-by-line comments
34 – change “Ebolavirus-mediated” to “Ebola virus disease.” Ensure “outbreak” and “epidemic” are used correctly. Check formatting throughout the paper to make sure of correct viral nomenclature usage (i.e. 56 – “ebolavirus” and “Marburgvirus”).
94/95 – remove the duplicate sentence.
161 – does this anti-filo antibody recognize SUDV and MARV? Were these also tested? If not, why?
Fig 1B – the distinction between constructs and mono mix controls could be better arranged/aligned.
218-20 – isn’t this stated earlier at 198? Unless at 198 the total mixture is 10^8, and at 218 this is increased to 3x10^8. Clarify.
224/Fig S1 – is this a separate mouse experiment? Were these single antigen infections, then each GP (rather than pool), or were they both? Clarify.
Fig 3 – why are the BALB/c mice reductions not significant for (gly) for EBOV and SUDV? By eye, this looks even more significantly different that those recorded for the CD-1 mice. Also, it would be beneficial to address the lower immunogenicity in some 2A or gly vaccinated mice compared to mono mix, as it seems to show high variability and thus implies potentially lower universal protection in some individuals.
236/238 – avoid “data not shown,” even for controls. What about MARV GP or CD-1 mouse data at 21 weeks?
256 – clarify in text that the responses are compared to mono mix with boost (and not just mono mix alone).
Fig 4C – “CD1” should be “CD-1”; fold-changes should have stats.
258 – clarify that this is referring to Fig 2’s data. Why were CD-1 mice not tested/shown in this experiment (the one described in Fig 2) as well? Is this humoral immunogenicity experiment data that wasn’t shown? Clarify; it might be clearer if the reference CD-1 data is shown somewhere.
274 – the phrasing is a little clunky here; make clearer and mention IL-2 here as well. Perhaps use “/” between cell marker targets. Is “peptide specific” the same as “peptide pools” as in M&M? This seems contradictory.
Fig 5a – EBOV GP with mono mix seems to be oddly variable, with most animals negative. Explain.
Fig 5b – this needs better clarification. There is no explanation in the text or legend of what these axes are. Also, why is there no bar for EBOV GP here, were no cells with these targets identified? If so, why might this be?
Fig S3 – it might be worth including the neutralization assay as a main figure.
301 – this sentence has one too many “or”, the phrasing is clunky.
302 – was this 5x10^8 IU for the mix total, or for each mono construct individually?
Fig 6 – “up to” 6 animals is vague, be specific. Are 2-3 logs of virus normal for these protected animals? If so, clarify based on previous studies. It would be better to set the axes to 0 rather than at 10^2. What is the limit of detection of the assay used, and how was viral load determined? Did the surviving GPs seroconvert? If so, provide this data.
323 – I’d suggest starting a new paragraph here related to clinical signs and maintaining a single paragraph through line 331.
335 – define “negative.” I’d revise this to “below limit of detection” if indeed 10^3 is below that; if not, revise to a more accurate descriptor.
338 – change to “endpoint” and capitalize “Figure 5.”
Author contributions – names here probably should be initialized rather than spelled out.
Reviewer 2 Report
This is a well-structured manuscript which discusses the creation of a single dose trivalent filo (EBOV, SUDV, MARV) vaccine using an Adeno vector. The authors clearly lay out the problem in the introduction - will surely become more important over the years now that we have single filo (EBOV) vaccines that have been shown to be effective - both the one dose rVSV-EBOV vaccine and the Ad-MVA boost 2 dose vaccine. While there has been some movement on development of a prime-boost vaccine for the three primary filo targets - there is a push for a single dose vaccine - that does not require follow-up. The authors present a potential candidate which expresses three primary filo targets, produces an antibody response for all three, and has been shown effective in guinea pigs who were vaccinated/exposed to EBOV. They have not yet challenged with the other two targets.
The methodology presented is appropriate for their primary question, but as they mention in the limitations/need for future research - there are additional tests that should be completed. They compare variations of their tri-filo vaccine to monovalent versions - and show similar ab responses. They present information on a number of variations as well and figures to support their conclusions.
I would suggest that in order to better help the reader understand the different variations, the authors should include a table with the outcomes.
In Figures 4 and 5a, the candidate vaccine triFilo(2a) which showed the most merit, had lower values (fold change, GP, and IFN-y), yet worked well in the challenge - could the authors address this in the discussion and what it could mean for challenge with the other viruses. There has been some work on the role of GP and generation of neutralizing antibodies - and if having neutralization antibodies is necessary for protection. It would be appreciated if the authors could also address this.
The writing was clear and concise - I have no grammatical edits.
Round 2
Reviewer 1 Report
The authors have done an admirable job editing, clarifying and expanding upon their manuscript per the previous comments. It is also understandable, given the current pandemic, that the authors would have difficulty during this time to add additional protection data for the other two filoviruses, which could unfairly delay dissemination of information that could prove invaluable to filovirus vaccine development. Therefore, I agree with the authors that the work as it currently stands is important and deserves attention within the scientific community for further investigation.
However, it is possible that certain passages/conclusions within the manuscript could be seen as an overstatement or over-simplification of the results, which are very promising but, in my view, somewhat preliminary. Thus, I think it's important for the authors to ensure that the overall message is represented in as specific and upfront a way as possible in the title, summary, introduction and discussion (expanding upon what is currently written), including a) that this work only shows protection for Ebola virus and not the other two filoviruses, b) that additional animal challenges inclusive of these other filoviruses (possibly pre-clinical in guinea pigs and perhaps even NHPs) might be needed before advancement to clinical studies, which, as far as I'm aware, appears to be the standard threshold that's been historically needed to advance previous candidate filovirus vaccines, and c) an expanded rationale for why these additional challenges were not pursued (lack of correlates of protection, etc.) and why the protection results as presented for Ebola virus alone is nevertheless of vital interest to the field and a critical first step (for instance, comparison to previous vaccine candidates, the licensed vaccine, etc). In other words, the caveats inherent to the study need to be better emphasized and explained, and any claims of protection limited to Ebola virus alone (for instance, the title is too vague and could be misinterpreted as suggesting that the vaccine protects against all three viruses).
To be clear, while I think that additional pre-clinical work will probably be needed prior to a Phase I trial, I do think that this present study addresses a real need in the field and I commend the authors for advancing a more practical vaccine with the desire to drive down costs and thus allow for greater availability of critical viral countermeasures available in poorer, at-risk regions.
Below are a few additional minor comments that the authors should address, listed in no particular order:
- Double check proper use of "Ebola virus" and "CD-1" throughout the manuscript, including in the suppl materials.
- The ethics statement is much appreciated, but I think that papers containing BSL-4 experiments need a biosafety statement as well.
- In the guinea pig experimental design section of the M&M, you specify that the ChAdOx1 vector is "biEBOV". Shouldn't this be the triFilo(2A) vector? Figure 6 legend and Suppl Fig 7 also make mention of "ChAdOx1 Ebola vaccine" rather than the 2A vaccine, please clarify.
- Figure 1b: the dotted lines are appreciated, but actually my previous comment was in relation to the number/letter labels at the top, some of which don't align clearly to their respective lane.
- It might be a good idea for the intro and/or discussion to explicitly mention that MVA vaccines are better as boost than prime given the manufacturing/scalability issues. It might also be good to be slightly more explicit about safety concerns of VSV vector in the intro within the newly-added passage referencing the other single-vector studies to bolster the rationale for the Ad-based vector and why an alternative to the multivalent VSV vector is warranted (without having to sacrifice the more extensive detail already included on this topic in the discussion). This way, readers have a clear and early understanding of why this new single vector vaccine is being developed despite the existence of these other two vaccine candidates.
- Line 345 reads as the start of a new paragraph, and should start the paragraph that currently begins on line 346.
- Were any of the assays performed for the mice experiments also performed during the guinea pig study? If so, having those as suppl figures would be beneficial to compare responses across animals.
- It might be a good idea to discuss the lack of neutralization for SUDV in the discussion, especially in light of the lack of protective data for this virus.
- The formatting/punctuation of the suppl figures needs correction.
- The histology suppl figure shows representative control and mono mix slides, but there's no histo from the 2A-vaccinated animals; as this is the most important condition, please add. Also, H&E staining images for the vaccinated animals as a comparison to control would be helpful.
Author Response
Reviewer 1 Round 2
Thus, I think it's important for the authors to ensure that the overall message is represented in as specific and upfront a way as possible in the title, summary, introduction and discussion (expanding upon what is currently written), including a) that this work only shows protection for Ebola virus and not the other two filoviruses, b) that additional animal challenges inclusive of these other filoviruses (possibly pre-clinical in guinea pigs and perhaps even NHPs) might be needed before advancement to clinical studies, which, as far as I'm aware, appears to be the standard threshold that's been historically needed to advance previous candidate filovirus vaccines,
We have now expanded on this topic in the discussion (penultimate paragraph). We have reworded the abstract and introduction to make it clear that Zaire ebolavirus challenge was performed.
and c) an expanded rationale for why these additional challenges were not pursued (lack of correlates of protection, etc.) and why the protection results as presented for Ebola virus alone is nevertheless of vital interest to the field and a critical first step (for instance, comparison to previous vaccine candidates, the licensed vaccine, etc).
We have added further detail on this in the discussion.
In other words, the caveats inherent to the study need to be better emphasized and explained, and any claims of protection limited to Ebola virus alone (for instance, the title is too vague and could be misinterpreted as suggesting that the vaccine protects against all three viruses).
We have inserted the word ‘candidate’ in the title to relate to the reader that this is not yet a proven vaccine, but rather one amongst several contenders, and still being assessed.
- Double check proper use of "Ebola virus" and "CD-1" throughout the manuscript, including in the suppl materials.
Both of these have now been corrected.
- The ethics statement is much appreciated, but I think that papers containing BSL-4 experiments need a biosafety statement as well.
We have added the following to the methods section:
All procedures involving infectious EBOV were performed in the Containment Level (CL) 4 facility at Public Health England (PHE) following standard laboratory procedures and risk assessments.
- In the guinea pig experimental design section of the M&M, you specify that the ChAdOx1 vector is "biEBOV". Shouldn't this be the triFilo(2A) vector? Figure 6 legend and Suppl Fig 7 also make mention of "ChAdOx1 Ebola vaccine" rather than the 2A vaccine, please clarify.
We apologise for the error. We have corrected the M&M section and figure legends.
- Figure 1b: the dotted lines are appreciated, but actually my previous comment was in relation to the number/letter labels at the top, some of which don't align clearly to their respective lane.
The numbers/labels at the top of the lanes have now been adjusted appropriately in Figure 1.
- It might be a good idea for the intro and/or discussion to explicitly mention that MVA vaccines are better as boost than prime given the manufacturing/scalability issues. It might also be good to be slightly more explicit about safety concerns of VSV vector in the intro within the newly-added passage referencing the other single-vector studies to bolster the rationale for the Ad-based vector and why an alternative to the multivalent VSV vector is warranted (without having to sacrifice the more extensive detail already included on this topic in the discussion). This way, readers have a clear and early understanding of why this new single vector vaccine is being developed despite the existence of these other two vaccine candidates.
We have followed this suggestion and added another sentence to this effect in the introduction.
- Line 345 reads as the start of a new paragraph, and should start the paragraph that currently begins on line 346.
This has been corrected.
- Were any of the assays performed for the mice experiments also performed during the guinea pig study? If so, having those as suppl figures would be beneficial to compare responses across animals.
Unfortunately none of the immunogenicity assays were performed in the guinea pig study, as this was purely a challenge study.
- It might be a good idea to discuss the lack of neutralization for SUDV in the discussion, especially in light of the lack of protective data for this virus.
A speculative comment has been inserted.
- The formatting/punctuation of the suppl figures needs correction
This has now been addressed.
- The histology suppl figure shows representative control and mono mix slides, but there's no histo from the 2A-vaccinated animals; as this is the most important condition, please add. Also, H&E staining images for the vaccinated animals as a comparison to control would be helpful.
We have updated Supplemental Figure 8 with further images which include those from triFilo-2A vaccinated guinea pigs. The images are different to those originally presented as the methodology for photographing slides has changed recently; therefore all newly taken photos have been used.